# Learning to Share: Selective Memory for Efficient Parallel Agentic Systems

**Joseph Fioresi** [1]   **Parth Parag Kulkarni** [* 1]   **Ashmal Vayani** [* 1]   **Song Wang** [1]   **Mubarak Shah** [1]
Project Page: https://joefioresi718.github.io/LTS_webpage/

## Abstract

Agentic systems solve complex tasks by coordinating multiple agents that iteratively reason, invoke tools, and exchange intermediate results. To improve robustness and solution quality, recent approaches deploy multiple agent teams running in parallel to explore diverse reasoning trajectories. However, parallel execution comes at a significant computational cost: when different teams independently reason about similar sub-problems or execute analogous steps, they repeatedly perform substantial overlapping computation. To address these limitations, in this paper, we propose **Learning to Share (LTS)**, a learned shared-memory mechanism for parallel agentic frameworks that enables selective cross-team information reuse while controlling context growth. LTS introduces a global memory bank accessible to all teams and a lightweight controller that decides whether intermediate agent steps should be added to memory or not. The controller is trained using stepwise reinforcement learning with usage-aware credit assignment, allowing it to identify information that is globally useful across parallel executions. Experiments on the AssistantBench and GAIA benchmarks show that LTS significantly reduces overall runtime while matching or improving task performance compared to memory-free parallel baselines, demonstrating that learned memory admission is an effective strategy for improving the efficiency of parallel agentic systems.

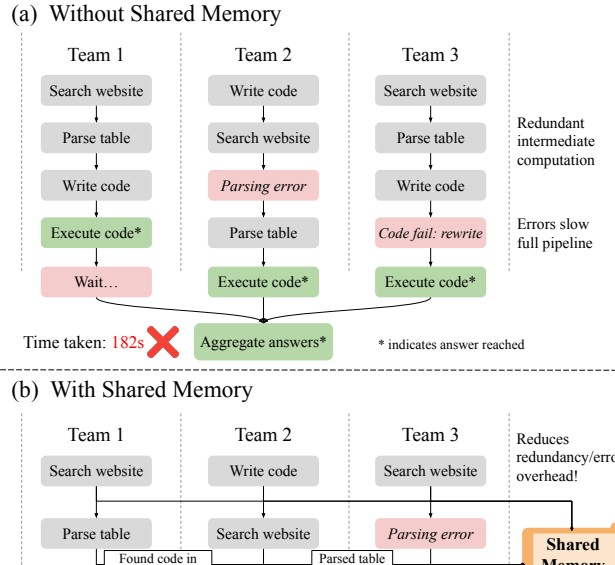

*Figure 1.* **Shared memory reduces redundant computation in parallel agentic execution.** Comparison of parallel agent teams solving a long-horizon task *without* (top) and *with* (bottom) shared memory. (a) Without shared memory, teams independently repeat overlapping intermediate steps (e.g., web search, table parsing, code writing), and errors in one branch propagate additional retries, increasing overall latency. (b) With a shared memory bank, teams reuse previously discovered intermediate results, avoiding redundant work and reducing error overhead. As a result, the system converges in fewer total steps and lower wall-clock time.

## 1. Introduction

Recent progress in large language models (LLMs) has led to the development of LLM-based agentic systems, in which

[1]Institute of Artificial Intelligence, University of Central Florida, Orlando, United States. Correspondence to: Joseph Fioresi <joseph.fioresi@ucf.edu>.

*Proceedings of the 43rd International Conference on Machine Learning*, Seoul, South Korea. PMLR 306, 2026. Copyright 2026 by the author(s).

a team of agents iteratively plans, reasons, and interacts with external tools or environments to solve complex tasks that extend beyond single-shot generation (Yao et al., 2022; Hong et al., 2023; Zhuge et al., 2024; Fourney et al., 2024; Du et al., 2023; Wang et al., 2024; 2025). These systems decompose problems into sequences of intermediate steps such as information retrieval, parsing, and code execution, enabling strong performance on long-horizon tasks that require sustained decision making. To further improve robustness and solution quality, recent work has proposed running multiple agent instances or teams in parallel, allowing the

system to explore diverse reasoning trajectories and mitigate the effect of individual failure modes (Ning et al., 2023; Kim et al., 2024; Zhang et al., 2025a). However, as illustrated in Figure 1(a), this parallelism introduces a key inefficiency. Independent teams frequently perform overlapping intermediate steps such as repeated web searches, table parsing, or code generation, as a result, errors in one branch trigger additional retries, which further increase redundant computation. Although parallel execution improves final-answer reliability, it does not prevent repeated reasoning during execution, leading to unnecessary increases in runtime.

A central reason for this inefficiency is that existing parallel agentic frameworks treat agent trajectories as independent by design (Zhang et al., 2025a). Intermediate results produced by one team, for example, when multiple teams independently extract the same fact from a webpage or derive equivalent intermediate code, are discarded once generated, even when they may be directly useful to other teams. As a result, parallel execution can amplify inefficiency on long-horizon tasks, where agents repeatedly rediscover shared sub-steps. Aggregation mechanisms that operate only on the final output improve solution quality but do not address redundancy during intermediate execution. This observation motivates a natural question: *Can parallel agentic systems retain the benefits of parallel exploration while reducing redundant computation to achieve faster convergence?*

In this work, we propose a shared-memory-based parallel agentic system named *Learning to Share (LTS)* to improve system efficiency by reducing redundant computation. We employ a global shared memory as a mechanism for information sharing across parallel agentic teams, as demonstrated in Figure 1(b). The memory bank stores intermediate agent steps as textual key–value pairs, where each entry consists of a concise natural-language summary that serves as a retrieval key and the corresponding raw agent output as its value. By exposing teams to the set of summaries and allowing selective retrieval of full contents when needed, the system enables cross-team reuse of intermediate results without forcing synchronization or unbounded context growth.

However, naively sharing all intermediate steps is neither efficient nor desirable. Many agent actions are incidental, redundant, or specific to a single reasoning path, for example, failed tool calls or partial code attempts. Indiscriminately storing them can clutter agent contexts and degrade performance. To address this challenge, we introduce a learned memory controller that decides whether a candidate agent step should be admitted to shared memory or not. The controller is implemented as a lightweight language model that emits a single binary decision token per step. Given that there is no ground truth for each memory admission decision, we are left with sparse supervision where feedback is only derived from downstream task success metrics. To

train this controller under these conditions, we design a stepwise reinforcement learning objective with usage-aware reward shaping, enabling the controller to identify which intermediate steps are globally useful across parallel teams while explicitly controlling memory growth.

We evaluate our approach on the GAIA (Mialon et al., 2023) and AssistantBench (Yoran et al., 2024) benchmarks, which feature long-horizon, tool-intensive tasks that naturally expose redundancy in parallel agentic execution. Across both benchmarks, our LTS method improves performance while drastically reducing wall-clock runtime compared to memory-free parallel baselines. Analysis studies show that naive memory sharing fails to achieve similar gains, highlighting the importance of learning when and what to share.

Our contributions are threefold:

- We identify the computation redundancy issue in parallel agentic systems and propose a global memory to improve efficiency in parallel agentic frameworks.

- We introduce LTS, an RL-based learning strategy with usage-aware reward shaping to train a lightweight controller that selectively shares intermediate information.

- We provide empirical evidence on GAIA and AssistantBench that learned shared memory improves the efficiency of parallel agentic execution without sacrificing solution quality on complex, multi-step tasks.

## 2. Related Works

**Agentic LLM systems.** LLM-based agentic systems solve complex tasks by iteratively planning, acting, and incorporating feedback from tools and environments. Early frameworks such as ReAct integrate reasoning traces with tool use in a single-agent loop (Yao et al., 2022), while subsequent approaches develop more structured single-team pipelines for multi-step problem solving, including program synthesis, web interaction, and multi-tool coordination (Hong et al., 2023; Li et al., 2023; Zhuge et al., 2024; Wu et al., 2024; Wang et al., 2024; Du et al., 2023; Fourney et al., 2024; Wang et al., 2025; Qian et al., 2023). These systems typically operate through long sequences of intermediate steps, often querying external tools or the web, which makes them well suited for long-horizon, tool-intensive benchmarks such as GAIA (Mialon et al., 2023) and web-based environments like AssistantBench (Yoran et al., 2024), WebArena (Zhou et al., 2023), and REAL (Garg et al., 2025). Despite their success, these systems execute as a single reasoning trajectory, making them sensitive to suboptimal intermediate decisions that shape all downstream steps.

**Parallel reasoning.** A line of work improves robustness by generating multiple reasoning trajectories and selecting

among them at the end. Repeated sampling has been shown to substantially improve reliability in code generation and reasoning tasks (Chen et al., 2022; Yao et al., 2023; Brown et al., 2024; Chen et al., 2024). This idea was carried into multi-agent frameworks by M1-Parallel, which runs multiple agent teams in parallel and aggregates their final outputs to improve robustness (Zhang et al., 2025a). Interestingly, they find that simply sampling multiple trajectories with identical prompts is most effective. However, this design treats agent trajectories as fully independent, leading to repeated discovery of the same intermediate steps. To mitigate this inefficiency, we introduce a principled mechanism for selective cross-team information sharing, enabling efficient reuse of intermediate knowledge while retaining the exploratory power of multi-trajectory execution.

**Memory in agentic systems.** Memory mechanisms in agentic systems have been studied primarily as persistent structures that support long-term interaction, personalization, and cumulative learning over episodes, including social and behavioral memory (Park et al., 2023; Sumers et al., 2023; Wang et al., 2023; Zhong et al., 2024; Jimenez Gutierrez et al., 2024; Tan et al., 2025; Zhang et al., 2025c; Chatterjee & Agarwal, 2025; Fang et al., 2025; Nan et al., 2025; Yan et al., 2025) as well as task-level memories that accumulate over problem-solving attempts (Shinn et al., 2023; Zhang et al., 2025b; Yuen et al., 2025; Xu et al., 2025). In contrast, we do not aim to build persistent or cross-episode memory. Instead, we study ephemeral shared memory instantiated per task and shared across parallel agent teams, isolating the role of memory in reducing redundant computation during parallel reasoning, rather than improving long-term adaptation or personalization.

**Baseline parallel agentic frameworks.** MagenticOne (Fourney et al., 2024) is a generalist multi-agent system designed for complex, open-ended tasks involving multi-step reasoning and tool use. It consists of a lead **orchestrator** agent and multiple specialized agents with distinct capabilities (e.g., web interaction, file navigation, code generation, and program execution). It executes *sequentially*: at each iteration, the orchestrator selects a specialized agent, assigns it a subtask, receives the agent's output, and updates its internal state before proceeding to the next iteration. In our setting, the orchestrator consumes memory items during each step and decides how such information should influence subsequent delegation decisions.

M1-Parallel (Zhang et al., 2025a) builds on MagenticOne by instantiating multiple independent MagenticOne-style teams and executing them in parallel. Each team runs its own sequential workflow, producing a candidate final solution. To combine the solutions produced by parallel teams, M1-Parallel uses an LLM-based aggregation strategy that selects

a final output based on the set of team solutions and the original query. We directly adopt this aggregation strategy without modification.

## 3. Method

We propose *Learning to Share* (LTS), a learned shared-memory mechanism for parallel agentic systems that reduces redundant computation while preserving or improving task performance (Figure 1). Our method augments existing parallel agentic frameworks with a global memory bank and a lightweight controller that selectively admits intermediate agent steps based on their expected downstream utility.

**Notation.** We formalize the parallel agentic execution using the following notation. Given an input task $x$, the system instantiates $K$ parallel agent teams indexed by $k \in \{1, \ldots, K\}$. Each team contains an orchestrator language model $\mathcal{O}_k$ that executes a sequential workflow. At step $t$, the orchestrator agent produces an action or delegation based on the task input and its team-specific trajectory history $h_{1:t-1}^k$, resulting in an observation or agent output that is appended to the trajectory. Teams execute independently until termination, each producing a candidate solution $y^k$. A fixed aggregation model $\mathcal{A}$ combines the set of candidate solutions $\{y^k\}_{k=1}^K$ with the original input $x$ to produce the final output $\hat{y}$, following (Zhang et al., 2025a). While execution is indexed by parallel team, the controller architecture and learning objective are identical across teams. For clarity and brevity, we omit the team index $k$ in subsequent sections unless it is required for disambiguation.

### 3.1. Global Shared Memory Bank

To enable cross-team information reuse without forcing synchronization or trajectory merging, we introduce a global shared memory bank $\mathcal{M}$ that stores textual key-value pairs. Each memory entry consists of a concise natural-language *summary*, $s_i$, (key) and the corresponding raw *agent output*, $o_i$ (value):

$$\mathcal{M} = \{(s_i, o_i)\}_{i=1}^{|\mathcal{M}|}. \tag{1}$$

At each time step $t$, an agent produces an output $o_t$ in response to an input $u_t$ (e.g., an orchestrator instruction). We additionally generate a short natural-language summary $s_t$ that describes the outcome of this step. If admitted, the pair $(s_t, o_t)$ is stored in the shared memory bank. To control context growth during execution, orchestrators do not directly consume all stored memory values. Instead, they are exposed to the memory bank through the set of summary keys $\{s_i\}$. An orchestrator agent may inspect these summaries and selectively choose a key to inject its value into its context when needed. This key-value design allows teams to efficiently access globally useful information while preventing unbounded context expansion.

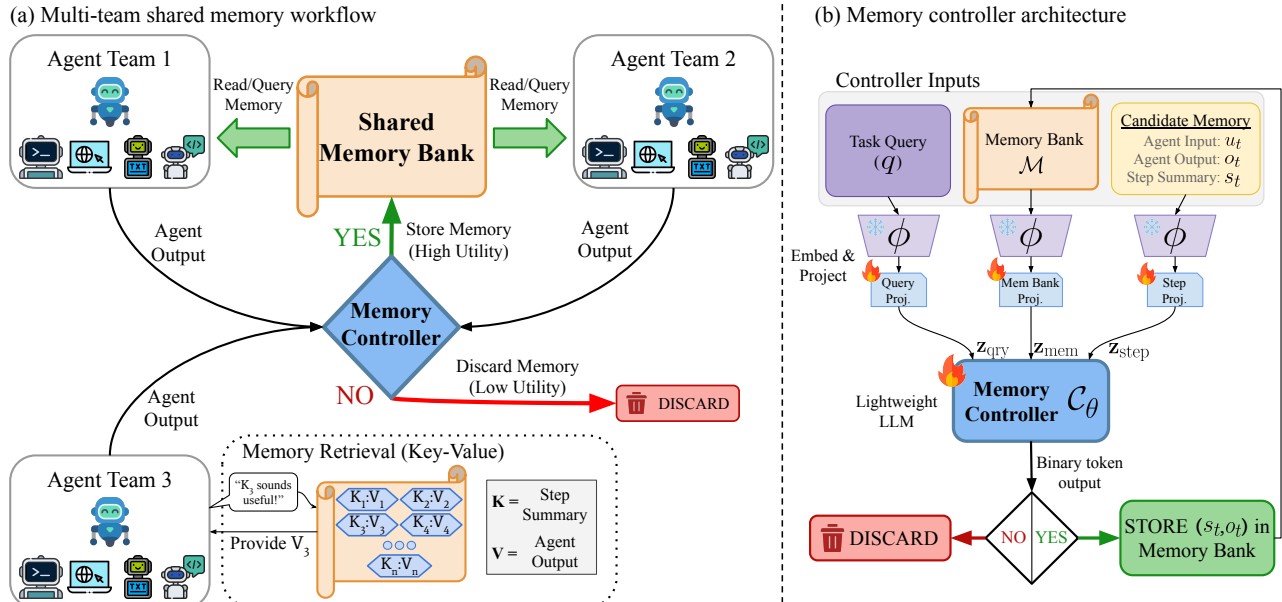

*Figure 2.* **Learning to Share: selective shared memory for parallel agentic systems. (a)** Parallel agent teams execute independently while interacting with a central *Shared Memory Bank*. After each agent step, a learned *Memory Controller* evaluates the intermediate result and selectively admits high-utility information into shared memory as a key-value pair (step summary, agent output) or discards it. Teams may query stored keys to reuse previously discovered results to reduce redundant computation (shown only for team 3, but all teams follow the same memory retrieval). **(b)** The memory controller receives embeddings of the task query, existing memory keys, and the current step (agent input, output, and summary) for context. These are projected into a shared token space and processed by a lightweight controller LLM, which emits a single binary decision indicating whether the step should be stored. Selective admission maintains a high-quality shared memory while accelerating convergence.

### 3.2. Learned Memory Admission

Not all intermediate agent steps are useful to other teams. Blindly sharing all steps can increase context length and introduce irrelevant information, harming both efficiency and accuracy. We therefore introduce a *memory controller* that learns to decide which steps should be added to the shared memory bank. Memory admission is formulated as a binary decision problem: for each candidate step $(u_t, s_t, o_t)$ produced by a team at time $t$, the controller decides whether the textual pair $(s_t, o_t)$ should be admitted to $\mathcal{M}$. The memory controller $\mathcal{C}_\theta$ is a lightweight language model that operates alongside the parallel agent teams. It observes the current execution context and emits a single binary decision token per step, indicating whether the step is globally useful enough to share.

**Controller inputs.** Effective memory admission requires reasoning not only about the current agent step, but also about the broader task context and the information already stored in memory. In particular, whether an intermediate step is worth sharing may depend on the original task query, the set of previously admitted summaries, and the relationship between the current agent input, output, and summary. To enable informed decisions, the controller is provided with all of this context rather than conditioning solely on the

current step. Concretely, at step $t$ we define the *controller context* $c_t$ as a sequence of token embeddings derived from multiple sources. All text is first embedded using a frozen text embedding model $\phi$. The controller context comprises: (i) an embedding of the task query $q$, given by $e^{\text{qry}} = \phi(q)$; (ii) embeddings of all existing memory keys (summaries),

$$E^{\text{mem}} = \big[\phi(s_1); \ldots; \phi(s_{|\mathcal{M}|})\big] ;$$

and (iii) embeddings of the current step triplet,

$$E_t^{step} = [\phi(u_t); \phi(s_t); \phi(o_t)] .$$

To interface with the memory controller language model, each component is mapped into the controller's input embedding space using lightweight learnable linear projection layers. We denote the resulting projected token sequences by $\mathbf{z}^{\text{qry}}$, $\mathbf{z}^{\text{mem}}$, and $\mathbf{z}_t^{\text{step}}$, respectively (e.g., $\mathbf{z}^{\text{qry}} = W_q(e^{\text{qry}})$). The controller context is then formed by concatenating these projected tokens:

$$c_t = \big[\mathbf{z}^{\text{qry}};\ \mathbf{z}^{\text{mem}};\ \mathbf{z}_t^{\text{step}}\big] . \tag{2}$$

Representing inputs as fixed-length embedded tokens rather than raw text keeps the controller lightweight and fast, while preserving the semantic context needed for informed memory admission decisions.

**Binary admission decision.** Given the controller context $c_t$, controller produces logits $\ell_t$ over a restricted set of valid tokens (e.g., YES / NO) at the final position:

$$\ell_t = \mathcal{C}_\theta(c_t), \qquad z_t \sim \text{Categorical}(\text{softmax}(\ell_t)), \quad (3)$$

where $z_t^k \in \{\text{YES}, \text{NO}\}$. If $z_t = \text{YES}$, the textual entry is admitted to the memory bank:

$$\mathcal{M} \leftarrow \mathcal{M} \cup \{(s_t, o_t)\}. \quad (4)$$

For efficiency, we maintain an internal cache of summary embeddings used by the controller; when a step is admitted, the embedding $\phi(s_t)$ is appended to this cache, while the memory bank itself stores only text. The controller emits exactly one decision per step and operates over a small output vocabulary, making its computational overhead negligible relative to orchestrator execution. A schematic of the memory controller is shown in Figure 2(b).

### 3.3. Training the Memory Controller

The utility of a memory admission decision is only observable through its downstream effect on the overall execution, leading to a challenging temporal credit assignment problem (Sutton et al., 1998). We therefore train the memory controller using a stepwise reinforcement learning objective (Williams, 1992) that combines group-relative advantage estimation (Gu et al., 2016; Schulman et al., 2017), usage-aware reward shaping, and explicit sparsity regularization. We model memory admission as a sequential decision process, where the controller policy $\pi_\theta(z_t \mid c_t)$ outputs a binary decision $z_t \in \{\text{YES}, \text{NO}\}$ at each agent step $t$. The controller context $c_t$ is formed from the task query, memory-key summaries, and the current step triplet (agent input, agent output, and step summary), as defined in Equation 2.

**Episode-level reward.** Each trajectory $\tau$, corresponds to a complete parallel execution for a single input and includes all agent steps, memory admission decisions, and the final aggregation. It receives an outcome-based reward $R(\tau)$ that reflects both system-level correctness and early convergence. Specifically, we define $R_{\text{agg}}(\tau) \in [0, 1]$ as the benchmark score of the final aggregated answer, and $R_{\text{first}}(\tau) \in [0, 1]$ as the benchmark score of the answer produced by the team that finishes first. The overall reward is given by:

$$R(\tau) = R_{\text{agg}}(\tau) + \lambda_{\text{first}} R_{\text{first}}(\tau), \quad (5)$$

where $\lambda_{\text{first}}$ controls the influence of first-team correctness. GAIA (Mialon et al., 2023) provides binary rewards, while AssistantBench (Yoran et al., 2024) yields partial credit.

**Group-relative advantage.** Absolute rewards are highly dependent on instance difficulty and exhibit high variance across inputs. To normalize for this effect, we adopt a group-relative baseline. For each input $x$, we sample a group

of $G$ independent executions $\{\tau^{(i)}\}_{i=1}^G$ under the current controller policy and compute a normalized base advantage:

$$A_{\text{base}}^{(i)} = \frac{R(\tau^{(i)}) - \mu_R}{\sigma_R + \epsilon}, \quad (6)$$

where $\mu_R$ and $\sigma_R$ are the mean and standard deviation of rewards within the group, and $\epsilon$ is a small constant for numerical stability. This formulation encourages the controller to prefer decisions that lead to relatively better outcomes for the same input rather than optimizing absolute reward magnitudes across heterogeneous tasks.

**Usage-aware shaping.** A successful episode does not imply that all memory admissions along the trajectory were useful. To provide denser supervision, we incorporate usage-aware reward shaping, motivated by (Ng et al., 1999; Schulman et al., 2015). For each trajectory $\tau^{(i)}$, we define the set of utilized memory indices $\mathcal{U}^{(i)}$ as the subset of admitted entries whose corresponding memory keys $s_j$ are selected by any orchestrator agent during execution. The stepwise advantage is then defined as:

$$\hat{A}_t^{(i)} = A_{\text{base}}^{(i)} + \beta \cdot \mathbb{I}\left(t \in \mathcal{U}^{(i)} \land R(\tau^{(i)}) > 0\right), \quad (7)$$

where $\beta$ controls the strength of the usage bonus. This shaping term assigns additional credit only to steps that demonstrably contribute to a non-zero task reward, mitigating spurious reinforcement of incidental decisions.

**Policy objective and sparsity.** We optimize the controller using a stepwise policy gradient objective (Williams, 1992; Sutton et al., 1998), weighted by the shaped advantage $\hat{A}_t$. The policy loss at step $t$ is defined as:

$$L_t^{\text{policy}}(\theta) = -\log \pi_\theta(z_t \mid c_t) \cdot \hat{A}_t. \quad (8)$$

To prevent degenerate "always admit" behavior, we introduce an explicit sparsity regularization term that penalizes the probability of admitting a step:

$$L_t^{\text{sparse}}(\theta) = \pi_\theta(z_t = \text{YES} \mid c_t). \quad (9)$$

The full optimization objective minimizes the expected sum of the policy loss and sparsity penalty:

$$\mathcal{L}(\theta) = \mathbb{E}_{\tau \sim \pi_\theta}\left[\sum_{t=1}^T \left(L_t^{\text{policy}}(\theta) + \lambda_{\text{sparse}} L_t^{\text{sparse}}(\theta)\right)\right], \quad (10)$$

where $\lambda_{\text{sparse}}$ controls the trade-off between utility and selectivity.

During training, all agents remain frozen. Gradients flow only through the projection layers and LoRA adapters of the memory controller, enabling stable learning under sparse outcome supervision.

## 3.4. Inference Workflow

At inference time, the system follows the standard parallel team execution, augmented with shared memory. Multiple teams execute in parallel, and after each agent step, the memory controller decides whether the step should be admitted to the global memory bank. Admitted entries expose lightweight summary keys to all orchestrators, with full memory values injected into context only when selected. Final outputs from all teams are combined using the LLM-based aggregation strategy (Zhang et al., 2025a).

## 4. Experiments

We evaluate the proposed shared-memory mechanism in parallel agentic systems with a focus on both task performance and computational efficiency. All experiments are designed to isolate the effect of shared memory and learned memory admission while fixing the underlying agentic framework.

### 4.1. Experimental Setup

**Benchmarks.** We evaluate our approach on two long-horizon agentic benchmarks designed to stress multi-step reasoning, tool use, and intermediate decision making: **GAIA** (Mialon et al., 2023) and **AssistantBench** (Yoran et al., 2024). Additional dataset details can be found in Appendix Section A.

**GAIA** consists of 165 tasks requiring iterative planning, external tool invocation, and the synthesis of intermediate results over extended execution traces. Tasks are organized into three difficulty levels (53 level 1, 86 level 2, and 26 level-3 tasks). We follow the standard GAIA evaluation protocol and report results on the official validation split.

**AssistantBench** evaluates web-based agents on realistic, multi-step tasks involving interaction with external tools and information sources. The benchmark contains 181 test tasks and supports graded partial credit, enabling more fine-grained assessment of task completion. We report results on the official test set of 181 tasks and train the memory controller exclusively on the 33-task development split to avoid test-set leakage. Notably, the controller is only trained on AssistantBench, even when evaluating on GAIA.

**Baselines.** MagenticOne (Fourney et al., 2024) represents the baseline with only one agent team ($K = 1$). For the rest of the experiments, we instantiate $K = 3$ parallel teams, consistent with prior work on parallel agentic execution (Zhang et al., 2025a). Each team consists of an orchestrator language model and a set of worker agents with access to the same tools and environment. We adopt the LLM-based aggregation strategy from M1-Parallel without modification, ensuring that any performance differences arise solely from the presence or absence of shared memory.

**Shared Memory Variants.** We compare the following variants to isolate the impact of shared memory and selective admission. `LTS-AddAll` inserts all agent steps into the shared memory bank, resulting in maximal sharing but unbounded context growth. `LTS-LLM` admits steps based on a prompted decision by a frozen LLM. `LTS` represents *Learning to Share*, which uses the learnable controller described in Section 3 to selectively admit memory entries based on their expected downstream utility. All variants share identical agents, tools, and aggregation procedures.

**Implementation Details.** We evaluate task performance and efficiency using both task success and wall-clock runtime. Task success is measured as the benchmark-defined score of the final aggregated answer. Wall-clock runtime is measured as the total elapsed time from task initialization to final answer aggregation, including all parallel agent execution, tool usage, and memory controller inference. All experiments are conducted on the same hardware configuration, using PyTorch (Paszke et al., 2019) on one NVIDIA H100 GPU. We build on the open-source Auto-Gen/MagenticOne framework[1] (Wu et al., 2024; Fourney et al., 2024). Unless otherwise stated, hyperparameters are held fixed across benchmarks, and the underlying agentic framework is identical across all compared methods. For evaluation, the system instantiates 3 parallel agent teams that execute synchronously. Each team is capped at a maximum of 30 agent steps to control execution length. All agents use either `gpt-5.1-2025-11-13` (Singh et al., 2025) or `Qwen3-32B` (Yang et al., 2025).

The proposed memory controller is implemented as a lightweight causal transformer based on the `Qwen3-0.6B` architecture with trainable LoRA (Hu et al., 2022) adapters (r=16, $\alpha$=16). Linear projection layers used to map embedded inputs into the controller's token space are trained jointly with the LoRA parameters. All textual inputs to the controller, including task queries, memory summaries, agent inputs, and agent outputs, are embedded using a frozen text embedding model $\phi$. In all experiments, $\phi$ is instantiated as the base `Qwen3-0.6B` *without* LoRA adapters for efficiency. To train the memory controller, we use the AssistantBench development set and collect multiple execution traces per task. No GAIA data is used for training. For each question, we sample 5 independent trajectories per epoch, capturing variability in agent behavior and downstream rewards. During training, we optimize the controller by minimizing Eq. 10 using AdamW (Loshchilov & Hutter, 2017). Controller decisions are sampled with a temperature of 1.2 to encourage exploration. For evaluation, we disable sampling and use greedy (argmax) decoding for all decisions. Further implementation details may be found in Appendix Section B.

---

[1] https://github.com/microsoft/autogen

*Table 1.* Results on the AssistantBench (Yoran et al., 2024) test set and the GAIA (Mialon et al., 2023) validation set, split by difficulty level. Runtime is shown as the average wall clock time taken per task. The number of parallel teams $K$ is 3. The proposed verified shared memory improves task performance while drastically reducing the runtime required to complete tasks.

| Method | Shared Memory | Model | AssistantBench | | | | | GAIA | | | | |
| | | | Easy Acc. | Med. Acc. | Hard Acc. | All Acc. | Execution Runtime ($\downarrow$) | Lvl 1 Acc. | Lvl 2 Acc. | Lvl 3 Acc. | All Acc. | Execution Runtime ($\downarrow$) |
|---|---|---|---|---|---|---|---|---|---|---|---|---|
| MagenticOne (Fourney et al., 2024) | ✗ | Qwen3 (32B) | 53.1 | 17.4 | 8.5 | 13.4 | 1084s | 41.5 | 16.3 | 3.8 | 22.5 | 758s |
| M1-Parallel (Zhang et al., 2025a) | ✗ | | 52.6 | 21.4 | 8.5 | 14.7 | 2239s | 43.4 | 18.6 | 7.7 | 24.8 | 1569s |
| **Learning to Share (Ours)** | ✓ | | **65.8** | **25.2** | **14.4** | **20.3** | 1479s ↓40.8% | **47.2** | **25.6** | **11.8** | **30.4** | 892s ↓55.0% |
| MagenticOne (Fourney et al., 2024) | ✗ | GPT-5.1 | 43.6 | 31.5 | 15.3 | 21.7 | 724s | 48.1 | 37.3 | 16.7 | 37.5 | 815s |
| M1-Parallel (Zhang et al., 2025a) | ✗ | | 57.3 | **35.2** | 16.0 | 24.0 | 1389s | 60.4 | 46.5 | **26.9** | 47.9 | 1005s |
| **Learning to Share (Ours)** | ✓ | | **61.0** | 35.1 | **20.0** | **26.7** | 882s ↓44.6% | **62.3** | **47.7** | **26.9** | **49.1** | 781s ↓25.1% |

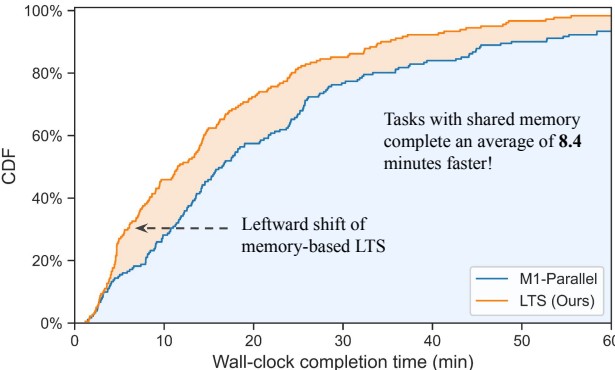

*Figure 3.* Cumulative distribution of wall-clock completion times on AssistantBench. Our `LTS` shared-memory approach shifts the runtime distribution left relative to memory-free M1-Parallel, indicating faster completion for a larger fraction of tasks. Selectively sharing intermediate results reduces redundant computation and lowers overall latency.

## 4.2. Results

We evaluate the proposed shared-memory mechanism in terms of both computational efficiency and task performance. Our results show that learned shared memory substantially reduces wall-clock runtime while improving task success relative to memory-free parallel baselines. Additional qualitative results are found in Appendix Section C.

**Runtime Reduction.** We first analyze wall-clock completion time on AssistantBench. Figure 3 shows a cumulative distribution function (CDF) plot of completion time for the baseline parallel agentic system and Learning to Share (`LTS`). Our shared-memory approach consistently completes tasks faster than M1-Parallel, despite the additional overhead of maintaining a memory. Compared to M1-Parallel without memory, shared memory reduces mean completion time by an average of at **8.4 minutes** and shifts the entire runtime distribution leftward, indicating faster convergence across tasks. Table 1 additionally shows the average runtime of each task in seconds.

*Table 2.* Comparison of shared memory admission strategies. We report task accuracy and average runtime (GPT-5.1). Naively sharing all intermediate steps (`LTS-AddAll`) reduces runtime but can hurt accuracy due to noisy memories. LLM-based filtering (`LTS-LLM`) partially mitigates this trade-off. The proposed selective memory (`LTS`) achieves the best overall balance, improving task accuracy while substantially reducing runtime.

| Memory System | AssistantBench | | GAIA | |
| | Acc. | Runtime | Acc. | Runtime |
|---|---|---|---|---|
| None | 24.0 | 1389s | 47.9 | 1005s |
| `LTS-AddAll` | 23.0 | **784s** | 44.2 | 967s |
| `LTS-LLM` | 25.7 | 856s | 45.8 | 853s |
| **`LTS` (Ours)** | **26.7** | 882s | **49.1** | **792s** |

**Task Performance.** Notably, the runtime improvements do not come at the cost of degraded solution quality. Table 1 shows that selective shared memory improves performance across both benchmarks and model backbones. Compared to the memory-free M1-Parallel, `LTS` achieves higher accuracy across nearly all difficulty levels while reducing runtime. On GAIA, `LTS` consistently improves performance across all difficulty tiers, with overall absolute gains of **+5.6 pp** for Qwen3-32B and **+1.2 pp** for GPT-5.1. A similar trend is seen with AssistantBench. These gains indicate that selectively sharing verified intermediate steps not only reduces redundant computation but also steers teams toward more reliable reasoning trajectories. Notably, the improvements are most pronounced on the hardest subsets of each benchmark, suggesting that Learning to Share is particularly effective for long-horizon tasks with many required steps/possible solution paths.

**Memory Admission Variants.** Table 2 compares different strategies for admitting intermediate steps into shared memory. All memory-enabled variants reduce overall runtime relative to the memory-free baseline, confirming that cross-team reuse of intermediate results improves execution efficiency. However, naively admitting all intermediate steps (`LTS-AddAll`) yields mixed performance, with reduced runtime but a drop in task accuracy, indicating that unfiltered memory can introduce irrelevant or misleading information

*Table 3.* Memory selectivity and utilization statistics for different admission strategies (AssistantBench, GPT-5.1). We report the percentage of steps admitted to memory, the memory recall rate, and the fraction of recalled memories used by a different team. Our proposed learnable variant LTS exhibits higher cross-team recall, indicating more generally useful shared memories.

| Method | Memories Saved (%) | Memory Recall (%) | Cross-Team Recall (%) |
|---|---|---|---|
| LTS-All | 100.0 | 25.8 | 66.0 |
| LTS-LLM | 44.4 | 18.0 | 66.6 |
| LTS (Ours) | 84.9 | 22.2 | **69.0** |

into team contexts. Filtering memory using a full LLM (LTS-LLM) improves accuracy over LTS-AddAll, but incurs additional computational overhead. In contrast, the proposed learned selective memory (LTS) achieves the best accuracy on both benchmarks while maintaining low runtime. These results highlight the importance of learned memory admission for balancing efficiency gains from shared memory with reliable task performance. This robustness arises from the usage-aware reinforcement learning objective, which explicitly rewards useful memory admissions that contribute to successful task completion, discouraging the retention of noisy steps.

**Memory Utilization Analysis.**   To better understand the effects of shared memory, we analyze memory selectivity and recall patterns in Table 3. Naively admitting all steps (LTS-AddAll) results in maximal memory growth but relatively low recall, indicating that many stored entries are never reused. In contrast, selective admission strategies substantially reduce the number of stored memories while maintaining comparable recall rates. Notably, LTS achieves the highest cross-team recall, suggesting that it preferentially admits intermediate results that are broadly useful across parallel teams. This behavior helps explain the improved efficiency of learned memory sharing by reducing redundant computation without overloading agent contexts.

### 4.3. Ablations and Analysis

**Method ablation.**   We conduct an ablation study on the proposed memory controller RL training objectives in Table 4. Removing usage-aware shaping leads to a noticeable increase in runtime and memory admissions, indicating that crediting only task-level success encourages the controller to over-admit steps that are not subsequently reused. Disabling sparsity regularization exacerbates this effect, resulting near-unconditional memory admission (akin to LTS-AddAll), the longest runtimes, and no gain in task accuracy. In contrast, the full objective (LTS) achieves the best balance between task performance, runtime, and memory selectivity, demonstrating that both usage-aware shaping and sparsity are necessary to learn efficient memory admission policies.

*Table 4.* Ablation study for RL training objectives on the GAIA validation set. Each component contributes to more efficient memory usage and improved overall performance.

| Variant | Task Acc. (%) | Runtime (s) | Memories Saved (%) |
|---|---|---|---|
| LTS | 49.1 | 792s | 84.9 |
| w/o Usage-Aware Shaping | 47.7 | 925s | 89.5 |
| w/o Sparsity Loss ($\lambda_{sparse} = 0$) | 47.4 | 983s | 98.9 |

*Table 5.* Scalability analysis with varying numbers of parallel teams $K$ on the GAIA benchmark. The same learned controller is used for all settings without retraining. LTS consistently improves both task accuracy and runtime relative to memory-free M1-Parallel across all tested team counts.

| Method | $K$ | Task Acc. (%) | Runtime (s) |
|---|---|---|---|
| M1-Parallel | 2 | 23.0 | 1161s |
| LTS | | **28.5** | **828s** |
| M1-Parallel | 5 | 25.4 | 2034s |
| LTS | | **31.6** | **1234s** |
| M1-Parallel | 10 | 22.4 | 3411s |
| LTS | | **29.1** | **2521s** |

**Scaling with parallel team count.**   We additionally evaluate LTS under varying numbers of parallel teams $K$ without retraining the controller. Results are shown in Table 5. Across all tested settings, selective shared memory consistently improves both task accuracy and runtime relative to memory-free M1-Parallel. For example, at $K = 5$, LTS improves GAIA accuracy from 25.4% to 31.6% while reducing average runtime from 2034s to 1234s. Even at $K = 10$, where memory growth and redundant exploration become substantially larger, LTS continues to provide clear gains over the baseline. These results suggest that the learned admission mechanism generalizes across different levels of parallelism and is not tightly coupled to a single team count. We additionally observe behavior consistent with prior findings from M1-Parallel (Zhang et al., 2025a), where excessively large team counts can eventually reduce overall performance despite increased parallel exploration.

*Table 6.* Robustness analysis for varying summary lengths on the GAIA validation set. The controller is trained only using short 15–20 word summaries and is evaluated without retraining on longer 45–50 word summaries. Performance remains substantially better than the memory-free baseline, indicating that the proposed approach is not highly sensitive to changes in summary length.

| Summary Length | Task Acc. (%) | Runtime (s) |
|---|---|---|
| No Summary (M1-Parallel) | 24.8 | 1569s |
| Short (15–20 words) | **30.4** | **892s** |
| Long (45–50 words) | 29.7 | 916s |

**Robustness to summary length.** Step summaries are generated using a stateless LLM with the same backbone as the underlying agentic framework. The summarization prompt requests a concise 15–20 word description of the purpose and outcome of each agent step to act as a memory bank key, consuming approximately 750 tokens per summary call for a typical 2000-character agent output. To evaluate robustness to summary quality and verbosity, we vary the target summary length from 15–20 words to 45–50 words without retraining the controller. Results are shown in Table 6. Both summary settings remain substantially better than the memory-free baseline, indicating that the proposed shared-memory mechanism is not highly sensitive to moderate changes in summary length. Short summaries perform slightly better than longer summaries, suggesting that concise summaries preserve the most salient information for downstream reuse while avoiding additional irrelevant detail. Although summary generation introduces additional computation, the resulting runtime remains significantly lower than the no-memory baseline due to reduced redundant execution across teams.

**Orchestrator-based memory admission.** We additionally evaluate a variant in which the orchestrator itself decides which intermediate results should be added to shared memory. Specifically, the orchestrator agent with its full context is prompted to admit or reject the most recent agent output. Results are shown in Table 7. Relative to the stateless `LTS-LLM` baseline, the orchestrator-controlled variant improves task accuracy, suggesting that access to trajectory-level context helps identify more useful intermediate information. However, it still underperforms the proposed `LTS` controller in both runtime and overall accuracy. We hypothesize that this occurs because the orchestrator is primarily optimized for solving its own trajectory, whereas the proposed controller is explicitly trained to predict cross-team downstream utility under a shared-memory budget. Additionally, the orchestrator accumulates substantially longer contexts during execution, therefore increasing the latency of repeated memory admission decisions.

*Table 7.* Comparison of LLM-based memory admission mechanisms on the GAIA validation set (Qwen3-32B). Orchestrator-based admission improves over a stateless LLM filter, but `LTS` achieves the best overall accuracy–runtime tradeoff.

| Memory Controller Variant | Task Acc. (%) | Runtime (s) |
|---|---|---|
| Stateless LLM | 26.8 | 928s |
| Orchestrator | 27.9 | 1108s |
| `LTS` | **30.4** | **892s** |

**Controller overhead.** A natural concern when introducing learned components into agentic systems is additional computational overhead. In practice, the proposed memory controller contributes negligibly to total execution time. We find that controller inference accounts for approximately **0.2%** of overall wall-clock runtime across tasks. This low cost stems from its lightweight design: the controller operates on short embedded representations, emits a single binary decision per step, and is implemented using a small `Qwen3-0.6B` backbone. As a result, the substantial runtime savings achieved by reducing redundant computation far outweigh the costs of memory implementation.

### 4.4. Scope and Limitations

The shared memory studied in this work is instantiated per task and does not persist across problem instances. As such, our approach does not aim to capture long-term personalization, user modeling, or cumulative knowledge acquisition; instead, it targets redundancy within a single parallel execution instance. Additionally, while our controller learns effective admission policies under sparse task-level supervision, it does not reason about memory deletion or revision within the existing memory bank. These design choices favor simplicity and efficiency, while leaving richer memory management mechanisms to future work.

## 5. Conclusion

We proposed a learned shared-memory mechanism for parallel agentic systems that enables selective reuse of intermediate information across teams. By introducing a global memory bank and a lightweight controller that learns which steps are worth sharing, our approach reduces redundant computation while matching or improving task performance. Experiments on the AssistantBench and GAIA benchmarks demonstrate consistent wall-clock runtime reductions compared to memory-free parallel baselines, whereas naive memory sharing fails to achieve similar gains. These results suggest that treating memory admission as a learned control problem is a promising direction for improving the efficiency of parallel agentic frameworks.

## Impact Statement

The primary goal of this paper is to advance the efficiency and accuracy of agentic machine learning systems. We do not foresee any immediate negative societal consequences unique to this contribution beyond those already associated with the deployment of large language model–based agents. Instead, the proposed approach may help make advanced agentic workflows more accessible and environmentally efficient, particularly in performance-critical settings where computational cost is a limiting factor.

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

## Appendix Overview

## A. Dataset Details

**GAIA (General AI Assistants) (Mialon et al., 2023).**   The GAIA benchmark is designed to evaluate the reasoning, planning, and tool-use capabilities of AI systems on real-world tasks that require multi-step execution and interaction with external information sources. Rather than narrowly scoped question–answer pairs, GAIA tasks require agents to invoke tools such as web search or document retrieval, synthesize intermediate findings, and combine evidence across multiple steps to arrive at a precise final answer. As a representative example, a GAIA task may ask: *"What was the actual enrollment count of the clinical trial on H. pylori in acne vulgaris patients from January–May 2018 as listed on the NIH website?"* Answering such a question requires locating the relevant clinical trial record, extracting the correct enrollment field, and verifying the time window before producing a factual response. The benchmark is organized into three difficulty tiers corresponding to increasing reasoning depth and tool usage, with 53 level-1 tasks, 86 level-2 tasks, and 26 level-3 tasks in the validation split used in this work. Level-1 tasks typically involve a small number of steps and limited tool interaction, while higher levels demand longer execution traces, multiple tool calls, and more complex information synthesis. Evaluation is performed via exact matching against a unique ground-truth answer for each task. GAIA was designed to be straightforward for humans yet challenging for current AI systems, highlighting persistent gaps in long-horizon reasoning and tool-based problem solving.

**AssistantBench (Yoran et al., 2024).**   AssistantBench evaluates the end-to-end capabilities of web-based agentic systems on realistic, long-horizon tasks that require planning, multi-step reasoning, and interaction with external tools and environments. Tasks are framed around practical user objectives, such as information gathering, comparison, and structured extraction from the web. For example, a typical query asks: *"Which gyms near Tompkins Square Park ($< 200m$) have fitness classes before 7am?"* Solving such tasks requires agents to issue search queries, navigate multiple websites, parse content, and integrate intermediate findings over extended execution traces. AssistantBench supports graded partial credit, enabling fine-grained evaluation of partial progress rather than relying solely on binary correctness. The benchmark contains 181 held-out test tasks, which we use exclusively for evaluation. To avoid test-set leakage, all memory controller training and hyperparameter selection are performed on the official 33-task development split.

## B. Implementation Details

**Memory Controller Architecture and Training Details.**   The memory controller is implemented as a lightweight causal transformer based on the `Qwen3-0.6B` architecture. All code is written using PyTorch (Paszke et al., 2019) and ran on a single NVIDIA H100 GPU. We fine-tune the controller using Low-Rank Adaptation (LoRA) with rank $r = 16$ and scaling factor $\alpha = 16$. All other backbone parameters remain frozen. In addition to the LoRA adapters, we train a small set of linear projection layers that map input embeddings into the controller's token embedding space. These projection layers are optimized jointly with the LoRA parameters. All textual inputs to the controller, including the task query, existing memory summaries, agent inputs, agent outputs, and step summaries, are first embedded using a frozen text embedding model $\phi$. In all experiments, $\phi$ is instantiated as the base `Qwen3-0.6B` model without LoRA adapters, which provides a shared semantic embedding space while keeping embedding computation inexpensive.

Controller training is performed exclusively on the AssistantBench development split. For each task, we sample $G = 5$ independent execution trajectories per training epoch under the current controller policy. This multi-trajectory sampling captures variability in agent behavior, execution paths, and downstream rewards, which is critical for stable policy optimization under sparse supervision. We train for 5 epochs, reusing the same trajectories 10 times within each epoch. During training, controller actions are sampled with a softmax temperature of 1.2 to encourage exploration and prevent premature collapse to overly conservative admission strategies. At evaluation time, sampling is disabled and all controller decisions are made via greedy (argmax) decoding. $\lambda_{\text{first}}$ is set to 1. All other components of the agentic system, including orchestrators, worker agents, and tools , remain frozen throughout training. Gradients flow only through the controller's LoRA adapters and projection layers. This design isolates learning to the memory admission policy, ensuring stable optimization and allowing improvements in efficiency and performance to be attributed solely to learned memory sharing behavior.

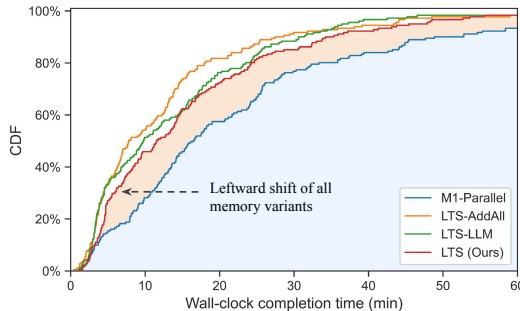

*Figure S1.* **Cumulative distribution of wall-clock completion times on AssistantBench for shared-memory variants.** All shared-memory variants shift the runtime distribution left relative to memory-free M1-Parallel, indicating reduced wall-clock latency due to cross-team reuse of intermediate results. Alternate admission strategies achieve larger runtime gains but exhibit lower task accuracy.

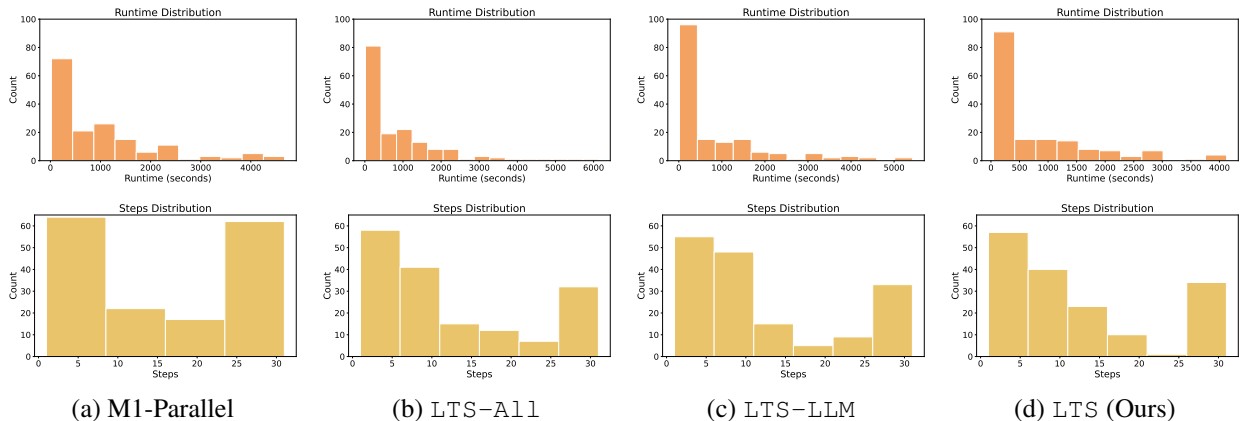

*Figure S2.* **Runtime and step-count distributions for shared-memory variants on AssistantBench.** Top row shows histograms of completion time per task, and bottom row shows the corresponding number of execution steps. Bin counts determined by Freedman–Diaconis rule. While all memory-enabled variants shift the runtime distribution toward shorter completion times, naive admission increases variance in step count and occasionally induces longer executions. In contrast, LTS achieves a consistent leftward shift in runtime while maintaining compact step-count distributions, indicating efficient reuse of intermediate results without introducing noisy or redundant steps.

# C. Additional Results

**Qualitative Results.** To provide intuition for the learned admission behavior, Figures S3 and S4 present qualitative examples of rejected and accepted memory entries, respectively. The rejected example shows an intermediate step that is locally useful but highly path-specific, offering little benefit for other parallel teams and thus being filtered out by the controller. In contrast, the accepted example illustrates a step that exposes reusable information needed by multiple teams, such as validated file contents and shared structural metadata.

**Further Runtime Analysis.** Figure S1 presents the wall-clock runtime distributions for alternative shared-memory admission strategies. While all variants reduce completion time relative to a memory-free version, naive admission policies achieve faster runtimes at the cost of reduced task accuracy. In particular, admitting all intermediate steps lowers latency but introduces noisy or misleading information that degrades performance. The learned selective memory strategy LTS achieves a favorable balance, retaining most of the runtime gains from shared memory while preserving high task performance.

Figure S2 provides a distributional view of execution behavior across shared-memory variants. Relative to the memory-free M1-Parallel (Figure S2a), all memory-enabled methods reduce wall-clock runtime by enabling reuse of intermediate results. However, naive memory admission (LTS-All) exhibits increased variability in step count, reflecting the accumulation of redundant or low-utility information that can prolong execution. LLM-based filtering (LTS-LLM) partially mitigates this effect but still introduces additional variance. In contrast, the proposed LTS method achieves both a pronounced leftward shift in runtime and a compact step-count distribution, indicating that learned selective admission effectively reduces redundant computation without increasing execution depth.

**Question:** According to the World Bank, which countries had gross savings of over 35% of GDP for every year in the period 2001-2010? Give your answer as a comma-separated list of countries in alphabetical order. Use the countries most common names in english when answering.

**Team 3 Orchestrator Agent** 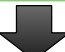 **(after step 3)**

**Output:** *Based on shared memory*, the World Bank data for indicator NY.GNS.ICTR.ZS has already been downloaded to `/workspace/API_NY.GNS.ICTR.ZS_DS2_en_csv_v2_174274.zip`. Please run the following Python code to process it:
```python
import zipfile
import pandas as pd
from pathlib import Path
...
print('Number of qualifying countries:', len(qualifying_sorted))
print('Comma-separated list:')
print(', '.join(qualifying_sorted))
```

Input to Computer Terminal Agent

**Team 3 | Step 4 : Computer Terminal Agent** 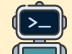

**Output:** The script ran, then exited with an error (POSIX exit code: 1)
Its output was:
Traceback (most recent call last):
  File "/workspace/tmp_code_4f26a10cba162de4127173a310e96ba343b20.py", line 21, in <module>
    df = pd.read_csv(main_csv)
         ^^^^^^^^^^^^^^^^^^^^
...
  File "pandas/_libs/parsers.pyx", line 891, in pandas._libs.parsers.TextReader._check_tokenize_status
  File "pandas/_libs/parsers.pyx", line 2061, in pandas._libs.parsers.raise_parser_error
pandas.errors.ParserError: Error tokenizing data. C error: Expected 3 fields in line 5, saw 70

**Summary:** Ran code to process World Bank data, resulted in pandas error.

Is step worth saving to memory?

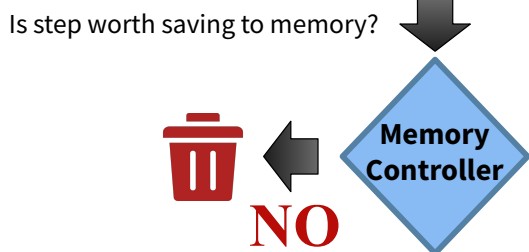

**Memory Controller**

NO

**Discarded:** Error step does not solve anything or find useful information; will add noise to shared memory. Learned memory controller successfully prevents from being saved.

*Figure S3.* **Example of a rejected memory entry.** Qualitative illustration of the learned memory controller rejecting an intermediate agent step. In this example, the agent executes code that fails due to a parsing error, producing no reusable intermediate result. The controller correctly identifies that it does not contribute useful information for other teams and prevents it from being added to the shared memory bank. This behavior helps maintain a high-quality memory by filtering error states and noisy artifacts that would otherwise increase context size and hinder downstream reasoning.

**Question:** Using the Biopython library in Python, parse the PDB file of the protein identified by the PDB ID 5wb7 from the RCSB Protein Data Bank. Calculate the distance between the first and second atoms as they are listed in the PDB file. Report the answer in Angstroms, rounded to the nearest picometer.

**Team 3 Orchestrator Agent** 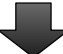 **(after step 1)**

**Output:** Please open the file `7dd30055-0198-452e-8c25-f73dbe27dcb8.pdb` in the current working directory and show the first 30–40 lines of its contents. I want to verify that it is a standard PDB file for PDB ID 5wb7 and confirm which lines correspond to the first and second atoms (the first two ATOM or HETATM records).

Input to File Surfer Agent

**Team 2 | Step 2 : File Surfer Agent** 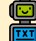

**Output:** HEADER    SIGNALING PROTEIN                28-JUN-17   5WB7
TITLE     CRYSTAL STRUCTURE OF THE EPIDERMAL GROWTH FACTOR RECEPTOR
TITLE    2 EXTRACELLULAR REGION IN COMPLEX WITH EPIREGULIN
COMPND    MOL_ID: 1;
COMPND   2 MOLECULE: EPIDERMAL GROWTH FACTOR RECEPTOR;
COMPND   3 CHAIN: A, B, C, D;
COMPND   4 SYNONYM: PROTO-ONCOGENE C-ERBB-1,RECEPTOR TYROSINE-PROTEIN KINASE
COMPND   5 ERBB-1;
COMPND   6 EC: 2.7.10.1;
COMPND   7 ENGINEERED: YES;
COMPND   8 MOL_ID: 2;
COMPND   9 MOLECULE: PROEPIREGULIN;
REVDAT   5 29-JUL-20 5WB7   1    COMPND REMARK HETNAM LINK
REVDAT   5 2           1    SITE  ATOM
REVDAT   4 04-DEC-19 5WB7   1    REMARK
REVDAT   3 01-NOV-17 5WB7   1    JRNL
…

**Summary:** Inspect EGFR-proepiregulin crystal structure metadata to identify complex components, origins, and experimental context

Is step worth saving to memory?

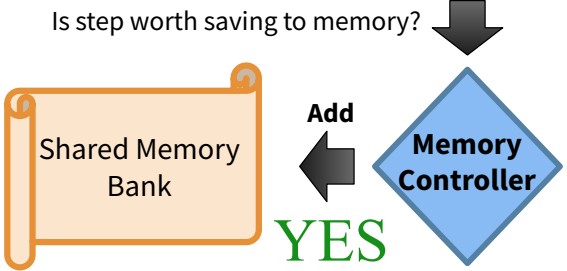

**Add**

**Memory Controller**

**Shared Memory Bank**

YES

**Saved:** Reading file is successful and finds useful information for solving the task. Learned memory controller decides to save to memory.

*Figure S4.* **Example of an accepted memory entry.** This example illustrates a case where the memory controller chooses to admit an intermediate step because it provides broadly useful information. The task asks for the distance between the first two atoms in the PDB structure for protein 5WB7, requiring agents to fetch, inspect, and parse the associated PDB file. Here, one team retrieves and displays the opening portion of the PDB file, confirming that the correct structure was loaded and exposing the ordering of ATOM records, which downstream agents need to compute the coordinate distance. The learned controller judges this step as globally useful: the retrieved file header and metadata help any team confirm file validity, avoid repeated downloads, and identifies the first atom in the structure. As shown in the figure, the controller outputs YES, admitting the summary and output into shared memory. This allows other teams to reuse the validated structure information directly, preventing repeated file inspections and accelerating progress toward the final solution.

