# OpenReview forum: "Learning to Share: Selective Memory for Efficient Parallel Agentic Systems"
_ICML.cc/2026/Conference — ICML 2026 regular_

### Official Review · Reviewer_jdaM · 2026-03-12

**Soundness:** 3
**Presentation:** 4
**Significance:** 4
**Originality:** 3
**Overall Recommendation:** 5
**Confidence:** 2

**Summary:**

This paper proposes Learning to Share, a method for improving efficiency in parallel LLM agent teams by selectively sharing useful intermediate results. LTS introduces a shared memory bank that stores step summaries and outputs produced by agents, along with a learned memory controller trained with reinforcement learning to decide which results are worth sharing. This allows parallel teams to reuse helpful intermediate information while avoiding noise from sharing everything. Experiments on GAIA and AssistantBench show that LTS reduces runtime and improves accuracy compared to memory-free parallel baselines, demonstrating that selective sharing can improve both efficiency and performance in multi-agent LLM workflows.

**Compliance With Llm Reviewing Policy:**

Affirmed.

**Key Questions For Authors:**

Q1. What would happen if you had the orchestrators themselves choose what to save in the share memory rather than a separate model? For example you could ask each orchestrator to distill its knowledge so far in the trajectory after each step and share it with the rest of the orchestrators. That would also avoid the context explosion.

**Limitations:**

yes

**Strengths And Weaknesses:**

**S1. LTS improves both task performance and execution time.**

That is a strong result, since many multi-agent coordination methods improve accuracy at the cost of extra latency or coordination overhead. The idea that selective sharing can simultaneously reduce redundant work and improve solution quality is both intuitive and practically valuable.

**S2. The core design is clean and well motivated.**

The shared memory bank plus learned controller gives a simple answer to both how and what multiple agents should communicate. The contrast with naive “share everything” baselines makes the contribution easy to understand.

**S3. The approach appears lightweight and broadly relevant.**

The coordination mechanism is small relative to the full workflow and duplicated reasoning and tool use are a real systems bottleneck.


**W1. The novelty is moderate at a high level.**

The underlying intuitiot that parallel agents can benefit from sharing useful intermediate results is natural, so the main contribution is the particular formulation of selective sharing rather than a fundamentally new paradigm for multi-agent reasoning.

**W2. The benefits may be workload-dependent.**

LTS seems most likely to help when parallel teams frequently rediscover overlapping intermediate results. It is less clear how much benefit remains when teams are more heterogeneous or when task structure offers less reusable overlap.

**W3. The learned controller adds some complexity and raises generalization questions.**

Even if inference-time overhead is low, the method still depends on an extra trained component, and the paper could do more to clarify how robust that component is across different tasks, agent configurations, and prompting styles.

---

> ### Author Rebuttal · Authors · 2026-03-31
>
> Thank you for the insightful review and for your constructive feedback. We address each of your questions below.
>
> >What would happen if you had the orchestrators themselves choose what to save in the share memory rather than a separate model? For example you could ask each orchestrator to distill its knowledge so far in the trajectory after each step and share it with the rest of the orchestrators.
>
> To answer this question, we have evaluated a variant in which the orchestrator decides what to store in shared memory, rather than LTS-LLM, which relies on a stateless LLM. See below Table A for results. This variant is competitive and beats out the stateless baseline in performance, but it still underperforms LTS in the final accuracy/runtime trade-off: on GAIA, it achieves **27.9** overall accuracy at **1108s**, compared to **30.4** accuracy at **892s** for LTS. We believe this happens because while the orchestrator has context for guiding the plan, it is largely focused on task solving within its own trajectory, whereas our LTS controller is trained specifically to predict *cross-team downstream utility* under a shared-memory budget. Note that the increased runtime is attributed to the orchestrator holding a lot of context tokens, slowing the inference speed of the constant memory decision. We thank the reviewer for this suggestion.
>
> **Table A: Evaluation of stateless LLM admission vs. orchestrator as memory manager. Results are shown on the GAIA benchmark (Qwen3-32B backbone).**
> |Method|Level 1|Level 2|Level 3|All Levels|Avg. Time|
> |-|:-:|:-:|:-:|:-:|:-:|
> |Stateless LLM|43.4|22.1|8.3|26.8|928s|
> |Orchestrator as Controller|43.4|23.2|11.8|27.9|1108s|
> |LTS|47.2|25.6|11.8|30.4|892s|
>
> > W1. The novelty is moderate at a high level.
> The underlying intuitiot that parallel agents can benefit from sharing useful intermediate results is natural, so the main contribution is the particular formulation of selective sharing rather than a fundamentally new paradigm for multi-agent reasoning.
>
> Our results show that while the high-level idea is intuitive, this intuition does **not** translate automatically into strong performance: a naive shared-memory approach does not immediately realize the full benefit of cross-team communication (Table 2, LTS-AddAll variant). To account for that, we introduce our novel and learnable memory controller mechanism tailored to inter-team communication and show through extensive experiments that it is effective in practice. Our contribution is in the concrete formulation and validation of our selective sharing mechanism for parallel agents.
>
> > W2. The benefits may be workload-dependent.
> LTS seems most likely to help when parallel teams frequently rediscover overlapping intermediate results. It is less clear how much benefit remains when teams are more heterogeneous or when task structure offers less reusable overlap.
>
> Our intended setting is aligned with prior work on **repeated-sampling parallelism**, where multiple trajectories improve robustness by exploring different solution paths and aggregating at the end [1]. The goal of LTS is to preserve the benefit of repeated sampling while reducing the redundant computation that arises when teams independently recover similar intermediate results. This is distinct from work that reduces latency by assigning **heterogeneous parallel subproblems** to different workers [2], which targets a different trade-off and is not the primary setting we study. Consistent with this positioning, M1-Parallel also examined heterogeneity within its framework and found that using heterogeneous plans was slightly less effective than repeated sampling with identical prompts. Our contribution is therefore aimed at maintaining the benefits of the repeated-sampling setting while improving efficiency.
>
> > W3. The learned controller adds some complexity and raises generalization questions.
> Even if inference-time overhead is low, the method still depends on an extra trained component, and the paper could do more to clarify how robust that component is across different tasks, agent configurations, and prompting styles.
>
> Our controller generalizes to numerous evaluation settings despite only being trained in one. Specifically, we train once using the GPT-5.1 backbone on the AssistantBench development set, then evaluate across all other settings; the GAIA benchmark, with a different backbone (Qwen3-32B), and now additional team size configurations ($K=2,5$). The gains persist across these changes, suggesting that the controller is learning a reasonably general notion of cross-team utility rather than overfitting to a single task or agent configuration.
>
> ### Citations:
> [1] Wang, Xuezhi, et al. "Self-consistency improves chain of thought reasoning in language models." arXiv preprint arXiv:2203.11171 (2022).
> [2] Ning, Xuefei, et al. "Skeleton-of-thought: Prompting llms for efficient parallel generation." The Twelfth International Conference on Learning Representations (2024).

---

> > ### Author Rebuttal · Reviewer_jdaM · 2026-04-03
> >
> > I appreciate the authors' reply and retain my high score of the paper.

---

> > > ### Author Response · Authors · 2026-04-07
> > >
> > > We thank the reviewer for retaining their positive evaluation of our work after our reply. We sincerely appreciate their time dedicated to reviewing our paper.

---

### Official Review · Reviewer_4CrQ · 2026-03-12

**Soundness:** 3
**Presentation:** 3
**Significance:** 3
**Originality:** 2
**Overall Recommendation:** 4
**Confidence:** 3

**Summary:**

This paper proposes the Learning to Share (LTS) mechanism to address the computational redundancy issue in parallel agentic systems, and designs a global shared memory bank plus a lightweight selective memory admission controller trained via reinforcement learning with usage-aware reward shaping and sparsity regularization. Experiments verify that this mechanism significantly reduces runtime while improving task accuracy, outperforming all baseline models by a notable margin.

**Compliance With Llm Reviewing Policy:**

Affirmed.

**Final Justification:**

The author resolved my questions, so I am maintaining my positive rating.

**Key Questions For Authors:**

see above.

**Limitations:**

yes

**Strengths And Weaknesses:**

**Strength**:
1. This paper addresses the computational redundancy in multi-agent collaboration and proposes a shared memory mechanism for mitigation, with highly meaningful research content.
2. The paper is well-organized with clear logic, and the research problem, motivation and methodology are presented in an intuitive manner.
3. Sufficient experiments are conducted on the shared memory mechanism, and the remarkable experimental results demonstrate its significant performance improvement.

**Weaknesses**:
1. Innovativeness: The core design integrates reinforcement learning-based admission control with the traditional centralized key-value shared memory, failing to break through the underlying framework of multi-agent shared memory and merely representing an optimized combination of existing technologies.
2. Experimental comparisons: The paper compares only two basic baselines and does not include other shared-memory-based methods. This limits the completeness and persuasiveness of the experimental evaluation.
3. Analysis of the team number $K$. The experiments fix $K=3$ without studying the impact of different $K$ values on the LTS mechanism. This analysis is necessary to evaluate the scalability of the approach and the trade-off between system efficiency and performance.
4. Evidence for redundancy reduction. Although the paper reports runtime and memory statistics, these metrics are indirect indicators of redundancy reduction. Providing explicit measurements of duplicated steps statistics would better support the claimed computational redundancy reducing.

---

> ### Author Rebuttal · Authors · 2026-03-31
>
> Thank you for the positive review, we respond to each of your points below.
>
> >Innovativeness: The core design integrates reinforcement learning-based admission control with the traditional centralized key-value shared memory, failing to break through the underlying framework of multi-agent shared memory and merely representing an optimized combination of existing technologies.
>
> To the best of our knowledge we are the first to present a shared-memory controller for parallel agent teams. Rather than maintaining memory *within* a single multi-agent team, we study memory *between parallel agent teams* that are independently exploring solution paths. This setting creates a distinct challenge of deciding which intermediate results are worth sharing across teams under limited context and runtime budgets, since naive sharing can easily overwhelm the benefit (Table 2, LTS-AddAll variant). Our RL-based admission controller is designed specifically for this cross-team setting, providing a practical mechanism for selective inter-team communication.
>
> > Experimental comparisons: The paper compares only two basic baselines and does not include other shared-memory-based methods. This limits the completeness and persuasiveness of the experimental evaluation.
>
>
> We now have compared our method with another memory system variant (see response to reviewer jdaM Q1-Table A). In addition to the memory-free baselines, our experiments already include multiple shared-memory comparisons in Table 2 that isolate the effect of different admission strategies. Our goal was to keep the memory mechanism itself simple and controlled so that the evaluation directly tests whether selective inter-team sharing is effective along with identifying whether a learned admission policy provides a better trade-off than heuristic alternatives. Most prior memory systems are designed for information sharing *within* a single agent or team rather than *between parallel teams*. Our intent was largely to show that even a simple shared-memory becomes highly effective when paired with learned admission in the cross-team setting.
>
> > Analysis of the team number $K$. The experiments fix $K=3$ without studying the impact of different values on the LTS mechanism. This analysis is necessary to evaluate the scalability of the approach and the trade-off between system efficiency and performance.
>
> We have now added experiments with $K=2$ and $K=5$ using the same setup as the main paper. See Table A below for results. LTS remains beneficial across all tested values of $K$: on GAIA, runtime is reduced from 1161s to 828s at $K=2$ and from 2034s to 1234s at $K=5$, all while improving task accuracy by 5.5 and 6.2 points, respectively. Importantly, the *same* controller continues to provide gains as $K$ changes, suggesting that the mechanism generalizes and is not tied to a single team count. Increasing $K$ increases parallel exploration, and we find that LTS helps mitigate the extra computational overlap that would otherwise accompany it.
>
> **Table A: Analysis on varying team number $K$.**
> |Method|$K$|GAIA|Runtime (s)|
> |-|:-:|:-:|:-:|
> |M1-Parallel|2|23.0|1161s|
> |Ours|2|28.5|828s|
> |M1-Parallel|5|25.4|2034s|
> |Ours|5|31.6|1234s|
>
>
> > Evidence for redundancy reduction. Although the paper reports runtime and memory statistics, these metrics are indirect indicators of redundancy reduction. Providing explicit measurements of duplicated steps statistics would better support the claimed computational redundancy reducing.
>
> To directly measure duplicated computation, we add an explicit step-level redundancy metric over GAIA logs: for each run, we extract agent-action steps from each parallel team, map each step to a short intent description (using a deterministic LLM prompt), and then count a step as redundant when a step with that intent is shared across multiple teams. See Table B below for quantitative results where the reported redundancy ratio is redundant_steps / total_agent_steps. Additionally, we count the total steps executed across teams and average over the $K$ teams to get the average number of steps taken to solve each task, showing that LTS reduces the number of steps required.
>
> **Table B: Overall per-problem average step counts on GAIA and AssistantBench (Qwen3 backbone).**
>
> |Method|GAIA (Num. Steps)|AssistantBench (Num. Steps)|Redundant Ratio|
> |-|:-:|:-:|:-:|
> |M1-Parallel|13.4|19.4|28.9%|
> |LTS|10.4|13.2|14.5%|

---

> > ### Author Rebuttal · Reviewer_4CrQ · 2026-04-01
> >
> > We appreciate the reviewer’s reply, which has resolved my questions.

---

> > > ### Author Response · Authors · 2026-04-07
> > >
> > > We thank the reviewer for confirming that our rebuttal addressed their concerns. We appreciate the time they dedicated to evaluating our manuscript.

---

### Official Review · Reviewer_iz1x · 2026-03-29

**Soundness:** 3
**Presentation:** 3
**Significance:** 3
**Originality:** 3
**Overall Recommendation:** 4
**Confidence:** 4

**Summary:**

This paper identifies the problem of redundant computation in parallel agentic systems, where independently running agent teams repeatedly perform overlapping intermediate steps (e.g., web searches, table parsing, code execution). To address this, the authors propose Learning to Share (LTS), a learned shared-memory mechanism. The controller is trained via stepwise reinforcement learning combining group-relative advantage estimation, usage-aware reward shaping, and sparsity regularization.

**Compliance With Llm Reviewing Policy:**

Affirmed.

**Final Justification:**

I maintain my initial recommendation of weak accept. The paper tackles a genuinely practical problem: redundant computation in parallel agentic systems, with a clean, well-designed shared-memory mechanism and a principled RL-based controller. The rebuttal partially addressed my main concern about scalability by providing new experiments, which show consistent accuracy and runtime gains. While the core contribution is technically sound, original, and clearly presented, the evaluation scope remains somewhat narrow. The partial new evidence strengthens my confidence but does not fully change my assessment.

**Key Questions For Authors:**

1. How does LTS perform as the number of parallel teams increases (e.g., K=5, 10)? Does the memory bank become noisy? Does the controller need retraining? Including even a small-scale study on this would significantly strengthen the paper.
2. How exactly are step summaries generated? What is the additional LLM cost? Have you measured sensitivity to summary quality (e.g., using shorter/longer or noisier summaries)? A positive answer on robustness would increase my confidence in the approach.

**Limitations:**

yes

**Strengths And Weaknesses:**

Strength

- The paper addresses a practical and well-motivated problem. Redundant computation in parallel agentic systems is a real bottleneck, and the observation that independent teams repeatedly perform overlapping intermediate steps is well-supported by the illustrative example in Figure 1. This is a timely contribution as parallel agentic frameworks gain traction.
- The overall system design is clean and well-thought-out. The key-value memory bank with summary-based retrieval is an elegant way to enable cross-team information sharing without forcing synchronization or unbounded context growth. The separation of summary keys (for browsing) and full outputs (for on-demand injection) is a practical design choice that balances informativeness with context efficiency.

Weakness

- All experiments use K=3 parallel teams with a cap of 30 steps per team. The paper does not investigate how LTS scales with more teams. With K=5 or K=10, the memory bank grows faster, retrieval becomes noisier, and the admission policy may need retraining. The scalability of the approach is an important open question left entirely unaddressed.
- The approach is evaluated exclusively within the MagenticOne/M1-Parallel framework using the AutoGen codebase. It is unclear how the method generalizes to other parallel agentic architectures with different orchestration patterns (e.g., tree-structured or graph-based execution rather than sequential within-team workflows).

---

> ### Author Rebuttal · Authors · 2026-03-31
>
> Thank you for taking the time and effort to provide a high-quality emergency review of our work. We address each of your questions below.
>
> >All experiments use K=3 parallel teams with a cap of 30 steps per team. The paper does not investigate how LTS scales with more teams. With K=5 or K=10, the memory bank grows faster, retrieval becomes noisier, and the admission policy may need retraining. The scalability of the approach is an important open question left entirely unaddressed.
> >
> >How does LTS perform as the number of parallel teams increases (e.g., K=5, 10)? Does the memory bank become noisy? Does the controller need retraining? Including even a small-scale study on this would significantly strengthen the paper.
>
> We have now added experiments with $K=2$ and $K=5$ using the same setup as the main paper. See Table A below for results. LTS remains beneficial across all tested values of $K$: for example, on GAIA, runtime is reduced from 1161s to 828s at $K=2$ and from 2034s to 1234s at $K=5$, all while improving task accuracy by 5.5 and 6.2 points, respectively. Importantly, the *same* controller continues to provide gains as $K$ changes, suggesting that the mechanism generalizes and is not tied to a single team count. Increasing $K$ increases parallel exploration, and we find that LTS helps mitigate the extra computational overlap that would otherwise accompany it. Note that we are currently running an experiment at $K=10$ and will share results as soon as we are able.
>
> **Table A: Analysis on varying team number $K$.**
> |Method|$K$|GAIA|Runtime (s)|
> |-|:-:|:-:|:-:|
> |M1-Parallel|2|23.0|1161s|
> |Ours|2|28.5|828s|
> |M1-Parallel|5|25.4|2034s|
> |Ours|5|31.6|1234s|
>
>
> >The approach is evaluated exclusively within the MagenticOne/M1-Parallel framework using the AutoGen codebase. It is unclear how the method generalizes to other parallel agentic architectures with different orchestration patterns (e.g., tree-structured or graph-based execution rather than sequential within-team workflows).
>
> We are currently running an additional experiment with an alternative execution structure to assess whether the benefit of selective cross-team sharing persists under a different orchestration pattern, and we will share those results during the discussion phase as soon as they are available.
>
> >How exactly are step summaries generated? What is the additional LLM cost? Have you measured sensitivity to summary quality (e.g., using shorter/longer or noisier summaries)? A positive answer on robustness would increase my confidence in the approach.
>
> Step summaries are generated from a stateless LLM with the same backbone as the framework. The cost would be around ~750 tokens based on a 2000 character output. The prompt used is exactly:
> >>"Summarize the agent step purpose and outcome in 15–20 words as a concise phrase, no quotes or trailing punctuation.
> Source: {source}
> Output: {content}"
>
> As such, we can directly influence the length of the summary by changing the word count. At this time, we are currently running an experiment to evaluate robustness with respect to the summary key and will share results as soon as possible.

---

> > ### Author Rebuttal · Reviewer_iz1x · 2026-04-01
> >
> > I appreciate the author's meticulous rebuttal and acknowledge it as a visionary and timely work. I will continue to maintain my positive attitude towards the author.

---

> > > ### Author Response · Authors · 2026-04-07
> > >
> > > Thank you for your kind remarks about our efforts! We provide the previously pending results below.
> > >
> > > >All experiments use K=3 parallel teams with a cap of 30 steps per team. The paper does not investigate how LTS scales with more teams. With K=5 or K=10, the memory bank grows faster, retrieval becomes noisier, and the admission policy may need retraining. The scalability of the approach is an important open question left entirely unaddressed.
> > > >
> > > >How does LTS perform as the number of parallel teams increases (e.g., K=5, 10)? Does the memory bank become noisy? Does the controller need retraining? Including even a small-scale study on this would significantly strengthen the paper.
> > >
> > > In addition to the $K=2$ and $K=5$ results shared above, we now include a $K=10$ experiment. Due to time/computational constraints, we evaluate $K=10$ on a randomly sampled half of GAIA; for direct comparison, we also report the corresponding $K=3$ result on the same subset. The controller is *not retrained* for this analysis. As shown in Table A2, LTS still improves over M1-Parallel at $K=10$, so the benefit of selective sharing persists as the number of teams grows. Note that these results are consistent with prior observations in the M1-Parallel paper that increasing the number of teams can eventually hurt performance.
> > >
> > > **Table A2: Analysis on varying team number $K$.**
> > > |Method|$K$|GAIA|Runtime (s)|
> > > |-|:-:|:-:|:-:|
> > > |M1-Parallel|3|24.7|1569s|
> > > |Ours|3|34.0|892s|
> > > |M1-Parallel|10|22.2|3508s|
> > > |Ours|10|28.4|2567s|
> > >
> > >
> > > >The approach is evaluated exclusively within the MagenticOne/M1-Parallel framework using the AutoGen codebase. It is unclear how the method generalizes to other parallel agentic architectures with different orchestration patterns (e.g., tree-structured or graph-based execution rather than sequential within-team workflows).
> > >
> > > To assess generalization beyond the MagenticOne/M1-Parallel sequential workflow, we implemented an additional experiment using a graph-based execution style running in parallel. Concretely, each team follows a directed workflow graph with a planner, tool-use specialist agent steps, another planning stage, more tool-use agent steps, then finally a final answer stage, rather than a sequential orchestrator-driven routing loop. We then add the same shared memory mechanism and memory controller on top of this alternate framework and evaluate on GAIA, shown below in Table B. The result is consistent with our main findings: introducing shared memory again improves task performance, indicating that the benefit of selective cross-team sharing is *not specific to sequential within-team execution* or to the original MagenticOne orchestration pattern. Notably, the same LTS memory controller is inferenced here with no additional training. Note that adding memory slightly increases runtime in the graph-based workflow since the fixed execution path leaves less opportunity to offset any memory cost through reduced steps.
> > >
> > > **Table B: Generalization to an alternate graph-based agent framework on GAIA.**
> > >
> > > |Method|GAIA|Runtime (s)|
> > > |:-|:-:|:-:|
> > > |Graph-based baseline|20.5|341s|
> > > |Graph-based + LTS|22.8|402s|
> > >
> > >
> > > >How exactly are step summaries generated? What is the additional LLM cost? Have you measured sensitivity to summary quality (e.g., using shorter/longer or noisier summaries)? A positive answer on robustness would increase my confidence in the approach.
> > >
> > > Since we can directly control summary length through the target word count, we evaluate robustness by increasing the summary budget from 15–20 words to 45–50 words. The controller is *not* re-trained for this analysis. As shown in Table C, both summary settings remain substantially better than the M1-Parallel baseline (No summary), indicating that the method is not highly sensitive to this change in summary length. Short summaries perform slightly better than long summaries, which we believe is because shorter summaries preserve only the most salient information while, whereas longer summaries introduce additional detail that is not always useful for admission decisions or downstream reuse. Regarding cost, summary generation accounts for approximately 22% of total runtime, but the overall runtime is still greatly reduced relative to the no-memory baseline.
> > >
> > > **Table C: Robustness analysis on varying summary lengths.**
> > > |Summary Length|GAIA|Runtime (s)|
> > > |-|:-:|:-:|
> > > |No summary (M1-Parallel)|24.8|1569s|
> > > |Short (15-20 words)|30.4|892s|
> > > |Long (45-50 words)|29.7|916s|

---

### Decision · Program_Chairs · 2026-04-30

**Decision:**

Accept (regular)

**Comment:**

This paper studies redundant computation in parallel agentic systems, where multiple teams independently repeat similar intermediate reasoning and tool-use steps, and proposes LTS. The method introduces a global shared memory bank together with a lightweight controller that decides which intermediate agent steps should be added to memory. Experiments on AssistantBench and GAIA show that LTS can substantially reduce overall runtime while matching or improving task performance.

The reviews generally agree that the paper addresses a timely and practically important problem, with a clean system design and convincing empirical results. The rebuttal further strengthened the submission by adding experiments on different numbers of teams, results on an alternative orchestration pattern, robustness analysis for summary design, and more direct evidence of redundancy reduction. These additions substantially addressed the main concerns about scalability, generalization, and the strength of empirical support.

There is still room to broaden the experimental scope, for example by including more shared-memory-related comparison methods and a wider range of agentic frameworks. However, given the importance of the problem, the clarity of the method, and the additional evidence provided during discussion, I believe the paper meets the bar for acceptance. I therefore recommend accept.